# Road side unit deployment optimization for the reliability of internet of vehicles based on information transmission model

**Jun Zhang**, **Guangtong Hu** *

College of Management and Engineering, Capital University of Economics and Business, Beijing, China

* charles3000@cueb.edu.cn

## Abstract

The Internet of Vehicles (IoV) makes it possible to transmit information in real time between vehicles, providing a modern approach for autonomous driving, traffic safety, and other applications. Roadside units (RSUs) contribute to the enhancement of IoV's reliability and transmission efficiency, while mitigating the impact of low IoV penetration. The objective of RSU deployment optimization is to minimize the total cost with the premise of ensuring IoV reliability. We construct a distance-based reliability measure for IoV, which is expressed as the proportion of information transmitted in online mode to the total transmission distance. The distance distribution of the online and offline transmissions is computed using the information transmission model. A bi-objective optimization model is established with the objectives of minimizing the cost of RSU and maximizing the reliability of IoV. Meanwhile, based on variable probabilities of crossover and mutation, a nondomination level-based NSGA-II (NNSGA-II) is designed to improve the solving efficiency. Numerical results show the advantage of the proposed model over the models evaluated with the objective of reducing transmission time can be up to 18% in different traffic scenarios, and NNSGA-II is significantly more computationally efficient.

**Data Availability Statement:** All excel files are available from the Mendeley database (DOI: 10. 17632/42y6rm36ns.1).

**Funding:** This study is supported by grants from the National Social Science Foundation of China

## 1. Introduction

Reliability holds an important status in applications of the Internet of Vehicles (IoV). IoV is of paramount significance in advancing Intelligent Transportation Systems (ITS) by enabling real-time data sharing, enhancing traffic management, and ultimately improving the efficiency and safety of transportation networks [1]. Plenty of attention has been given to IoV routing protocol [2–6] and applications [7–9], most research assumes that it is conducted under a reliable IoV environment. Road side unit (RSU) is a common approach to improving the reliability and transmission efficiency of IoV. However, RSUs cannot be deployed without limit considering the high cost of RSU deployment and maintenance [10]. To support the normal operation of IoV applications, this paper provides a reliable and economical method achieved through the optimization of RSU deployment.

In an IoV environment, Onboard Units (OBUs) collect some essential data about the vehicle, such as speed, position, direction, and acceleration speed, then transmit the information

(grant number 20BGL001), which is required to be unique. The funders had no role in study design, data collection and analysis, decision to publish, or preparation of the manuscript.

through vehicle-to-vehicle (V2V) or vehicle-to-RSU (V2R) [11]. The information transmission model describes the V2V transmission process of information of IoV. Most information transmission models primarily consider multi-hop transmission conducted through wireless networks [12–16]. However, when there is no connected vehicle (CV) within the communication range, information moves along with the vehicles, which is called ferry transmission [17]. This paper divides the mode of information transmission into online and offline, which represent the information transmit by multi-hop and move with the vehicle, respectively. Reliability is the probability that the system will perfectly perform its intended function over the entire mission time [18]. Nevertheless, in the online mode, information is transmitted among vehicles at speeds approaching the speed of light using radio waves, while in the offline mode, information moves alongside vehicles at the speed of their travel. Therefore, the time required for online transmission is far less than offline transmission in the process of IoV information transmission, the time-based reliability measure loses its reference value. To accommodate the practical needs of IoV, we propose an IoV information transmission model-based reliability measure, which is represented as the proportion of the distance of the information transmitted in online mode to the total distance transmitted.

Building upon the aforementioned perspective, we have developed a bi-objective optimization model aimed at reducing both the RSU cost and the penalties associated with online transmission distances falling below a specific threshold. To identify the optimal RSU deployment, the model is solved using the NNSGA-II (non-dominant NSGA-II) algorithm. The IoV environment and testbed are shown in Fig 1, and the distance transmitted via V2R is treated as the online transmission distance.

The proposed model demonstrates significant advantages in IoV environment dominated by RSUs. In scenarios with a high penetration rate of in-vehicle CVs, the performance is comparable to that of benchmark models. Additionally, NNSGA-II algorithm exhibits faster convergence speed and higher computational efficiency across various scenarios, validating its effectiveness in addressing RSU deployment optimization problems. An efficient RSU deployment reduces traffic accidents and associated social healthcare costs, enhances traffic efficiency, decreases congestion, saves time and fuel consumption, promotes logistics and commercial activities, and drives economic growth. Additionally, it lowers carbon emissions, protects the environment, improves traffic safety, and reduces personal injuries and property damage. In summary, the optimized RSU deployment strategy not only brings significant economic benefits but also generates positive social impacts.

The main contributions of this paper can be summarized as follows:

- A distance-based reliability measure is proposed to build the bi-objective optimization model for RSU deployment based on the IoV information transmission model. Compared to the prevalent RSU deployment optimization models, which primarily aim to reduce information transmission time, our proposed model, based on the distance-based reliability measure, offers a more accurate representation of the information transmission process within IoV.

- Considering both the transmission range of CV and RSU, we have extended the selection range of OD (Origin-Destination) points from the endpoints to encompass every location along the road, thereby extending the traffic scenario infinitely. Our work offers a more comprehensive perspective on IoV in the context of CV and RSU deployments. Before incorporating communication ranges, it is assumed that a single RSU could cover the entire road segment, offering only two deployment options for each road segment: deploy RSU or not. However, with the inclusion of communication ranges, the situation becomes more intricate. In the case study presented in this paper, for instance, five RSUs are needed to cover the entire road segment, expanding the RSU deployment options to six.

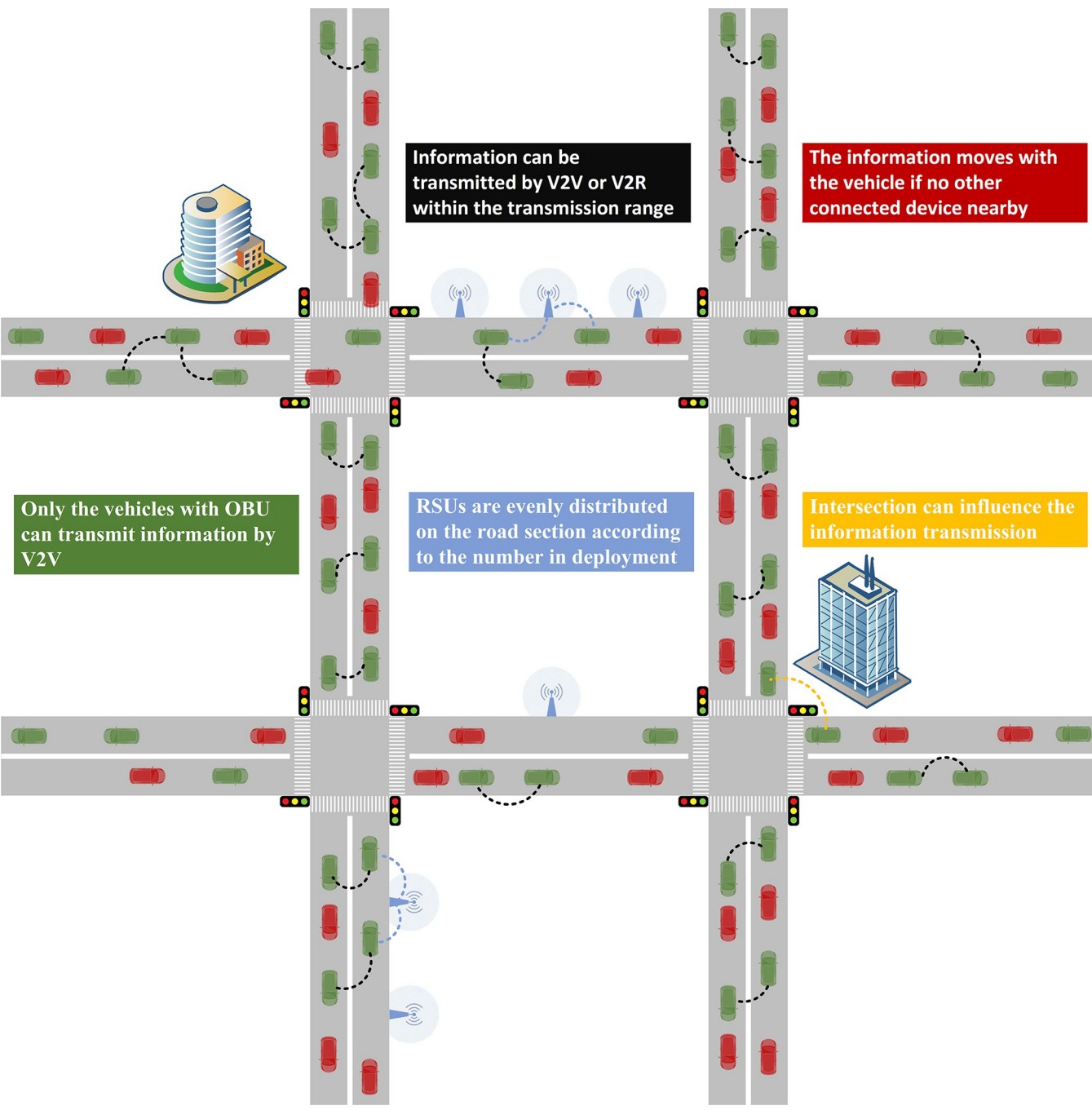

**Fig 1. Illustration of transmission model and testbed.**

- Introducing NNSGA-II, an algorithm based on NSGA-II and customized to handle the complex model presented in this study, all while achieving significantly improved convergence speed. Due to the expansion of the traffic scenarios and the inclusion of communication ranges in our consideration, the complexity of the model has been significantly elevated. NNSGA-II plays a pivotal role in tackling the complex optimization tasks posed by our model.

The structure of this paper is organized as follows. Section 2 discusses the related works. Section 3 describes the RSU optimization model based on the information transmission model. In Section 4, we propose a bi-objective optimization algorithm, NNSGA-II. In Section 5, we perform numerical simulations of the model solutions. Finally, the conclusions and prospects are presented in Section 6.

## 2. Related works

IoV applications heavily rely on advanced computing facilities and reliable networks. Hence, RSUs play a vital role in bolstering the reliability of IoV, a critical element for the seamless operation of IoV applications. Given the significant expenses associated with deploying and maintaining RSUs, it is imperative to minimize their number while ensuring the reliability of IoV.

Finding the optimal RSU deployment is an NP-hard problem, and many research studies have been carried out on this topic. Some studies use linear or non-linear programming methods, while others employ heuristic algorithms or develop mathematical models and optimization techniques.

For instance, Liang et al. in [19] modeled the RSU deployment question as a two-stage stochastic integer nonlinear programming problem with the objective of minimizing the combined costs of RSU and communication delay penalties. Considering joining the randomness of strategy selection of autonomous vehicles at intersections. Wu et al. in [20] established an information transmission model containing two RSU communication modes with considering the effects of wireless interference, vehicle distribution, and speed, which is solved by an integer linear programming model aiming at throughput maximization.

On the other hand, heuristic algorithms have also been widely explored. Gao et al. in [21] introduced OptGreDyn, an optimization algorithm that combines the greedy algorithm with dynamic programming to examine the optimal solution properties of a one-dimensional RSU deployment problem in a non-linear road environment. Magsino et al. in [22] proposed an EISHA-RSU deployment that strategically deploys RSUs in effective positions to maximize the sharing of environmental information among CVs, achieving higher connectivity and efficiency with fewer RSUs than existing deployment schemes. Silva et al. in [23] developed a Gamma-Reload-Deployment (GRD) strategy for RSU deployment based on V2R and V2V transmission efficiency, the results show that the GRD strategy can provide similar quality of service with fewer RSUs than the full coverage strategy. Mao et al. in [24] proposed an RSU deployment strategy, which takes vehicular transmission demand into account and utilizes an evolutionary feature-based algorithm to optimize RSU placement and boost the efficiency and robustness of the IoV. This scheme achieved lower delay and higher coverage time ratio than existing approaches in both urban and suburban road networks. Guerna et al. in [25] presented a novel ant colony optimization system to tackle the RSU deployment problem, aiming to increase network connectivity and reduce RSU crossing, the optimal RSU deployment scheme was searched by simulating the foraging behavior of real ant colonies.

Furthermore, some studies have focused on developing mathematical models and optimization techniques. Salari et al. in [26] developed mathematical models to optimize the deployment of RSUs and automatic vehicle identification (AVI) sensors for path flow reconstruction in CV environments, demonstrating that RSUs can provide more unique path flow information and require fewer sensors than AVI sensors in a mixed traffic environment. Moura et al. in [27] proposed a road network map that is based on middle centrality, solved by a simple genetic algorithm, and compared with other strategies on five real data sets.

**Table 1. The limitation of exist research the improvement of the proposed model.**

| Method | Literature | Limitation | Improvement |
|---|---|---|---|
| Integer Programming | Liang et al. (19), Wu et al. [20] | The complexity of the model is high, and the assumptions required are quite idealized. | This paper introduces a distance-based reliability metric, making the model more realistic and reducing bias from idealized assumptions. |
| Heuristic Algorithms | Gao et al. [21], Magsino et al. [22], Silva et al. [23], Mao et al. [24], Guerna et al. [25] | The effectiveness of these algorithms depends on specific scenarios and parameter settings, potentially performing poorly in other contexts. | This paper introduces a penalty cost when the proportion of online transmission distance falls below a certain threshold, thereby increasing the total system cost and encouraging the system to enhance online transmission ratios, thus mitigating the issue of heuristic algorithms failing to find a global optimum. |
| Mathematical Modeling and Optimization Algorithms | Salari et al. [26], Moura et al. [27] | There is a degree of idealization present, which might make it challenging to fully represent the complexities of the real world. | This paper elaborately describes how Roadside Units (RSUs) convert offline transmission to online transmission under different conditions, making the model more concrete and easier to understand and implement. This overcomes the issue of idealized assumptions in mathematical modeling and enhances the practicality and operability of the model. |
| Hybrid Management Strategies | Ni et al. [28] | The optimization decision granularity and coupling relationships between RSUs increase complexity. The effectiveness of the strategy might be constrained by specific traffic flow. | By incorporating a distance-based reliability metric, optimizing the proportion of online transmission distance, and detailing the RSU conversion function, the aforementioned limitations are addressed, enhancing the reliability and flexibility of RSU deployment. |

Additionally, Ni et al. in [28] proposed a cyber twin-based architecture for RSU management on the IoV, utilizing both static and mobile RSUs and a hybrid management strategy to meet V2X service demands while optimizing resource utilization.

Although existing research has made significant contributions to RSU deployment optimization, there is still a lack of research on the reliability analysis of IoV based on the information transmission model. More importantly, previous studies have rarely considered the distance-based reliability measure, which does not match the real-world IoV environment.

Compared with previous research, we introduce a distance-based reliability measure to more accurately reflect information transmission reliability in real-world IoV environments. Table 1 summarizes the previous research, highlighting their respective limitations and the improvements introduced in this study. We propose a new model that incorporates a penalty cost when the proportion of online transmission distances falls below a certain threshold, thereby optimizing information transmission reliability. When the proportion of online information transmission is too low, it increases the total system cost, prompting the system to seek ways to enhance the online transmission ratio. Additionally, we detail how RSUs convert offline transmissions to online transmissions under different conditions, making the model more concrete and easier to understand and implement.

## 3. Model

This section models the RSU deployment optimization problem as a bi-objective optimization programming based on the information transmission model. The components of the model on the IoV also will be introduced in this section.

### 3.1 Notations

The notation used in this paper is shown in Table 2.

RSU deployment is a complex problem, considering both IoV reliability and RSU cost, which can be modeled using the bi-objective optimization modeling. The RSU cost can be

**Table 2. Notations.**

| Sets | Describe |
|---|---|
| I | the set of roads ends |
| $\Omega$ | the set of scenarios |
| $\emptyset$ | the set of selected OD pairs |
| **Indexes** | |
| i | the index of road ends, $i \in I$ |
| $\omega$ | the index of scenarios |
| $\varphi$ | the index of origin in OD |
| **Decision Variables** | |
| $n_{i_1 i_2}$ | the number of RSUs on the link $(i_1 i_2)$ |
| $s_{i_1 i_2}^{\varphi}(\omega)$ | $s_{i_1 i_2}^{\varphi}(\omega) = 1$ represents the link $(i_1 i_2)$ is on the shortest path between the OD pair $\varphi$ in scenario $\omega$; otherwise, $s_{i_1 i_2}^{\varphi}(\omega) = 0$ |
| $c_{\omega}$ | the number of the cost that online transmission distance ratio below an acceptable threshold between all OD pairs |
| **Parameter** | |
| $\theta$ | the penetrance of CV |
| r | the communication range of CV |
| R | the communication range of RSU |
| $n_m$ | the maximum number of RSUs that can be deployed on the link |
| $L_o$ | the origin point of the road |
| $L_d$ | the destination point of the road |
| $L_s$ | the length of the shortest path of the OD pairs |
| $L_r$ | the length of the link |
| m | the number of the observed headways |
| $\hat{\sigma}$ | the estimator of the standard deviation of the Probability Density Function of the headways |
| K | the Gaussian kernel function |
| X | the observed headways when penetration is 1 |
| $X_{\theta}$ | the observed headways when penetration is $\theta$ |
| $\bar{v}$ | the average speed of the observed vehicle |
| $\bar{k}$ | the average hops in a single online transmission |
| $\tau_0$ | the maximum time for discovering the next CV |
| $\gamma$ | the threshold of online transmission ratio |
| $D_t(\varphi)$ | the total distance of information transmission in $\varphi$ |
| $D_o(\varphi)$ | the distance of information online transmission in $\varphi$ |
| $F_c$ | the converted online transmission distance |

easily obtained, but the IoV reliability needs to be calculated by the information transmission model. We model the information transmission in IoV based on the headway distribution function, and extract intrinsic patterns from the recorded headway data to describe the distribution properties of the headways. Different traffic conditions and penetration of CVs will lead to different information transmission modes, and the distance distribution can be calculated. With the distance distribution of online and offline modes, we develop a bi-objective optimization model that considers two optimization objectives: the penalty cost associated with the online distance ratio and the number of RSUs. It should be noted that for the convenience of calculation, the IoV reliability is expressed as the number that the online distance ratios below the specified threshold, details are shown in section 3.5.

## 3.2 Objective functions

The two objective functions of the bi-objective model are given by Eqs (1) and (2), with Eq (1) targeting the reduction of the number of RSUs and the costs associated with their deployment and maintenance, while Eq (2) aims to minimize the penalty cost arising from online transmission distance falling below a threshold for all OD pairs. Due to the stochastic nature of road traffic, the shortest path for information transmission and the minimum penalty cost will shift with the distribution of vehicles. Therefore, the model selects the RSU deployment that minimizes both objectives for all traffic conditions.

$$min \frac{1}{2} \sum_{i_1 \in I} \sum_{i_2 \in I} n_{i_1 i_2} \tag{1}$$

$$min \ c_\omega \tag{2}$$

Eqs (3)–(16) are the constraints of the proposed model.

## 3.3 Basic constraints

This sub-section describes some basic constraints on the model. Eq (3) restricts the number of RSUs on a single road section, so $n_{i_1 i_2}$ is a nonnegative continuous integer. The maximum value of n is obtained by Eq (4), which denotes the minimum number required by RSU to cover the entire link.

$$n_{i_1 i_2} \in \{0, 1, \ldots, n_m\}, \forall (i_1, i_2) \in I \tag{3}$$

$$n_m = \frac{L_r}{2 \times R} \tag{4}$$

In this model, the origin and destination points are not constrained to be located only at the endpoints of the road; they are allowed to be positioned at any point along the road. Eqs (5) and (6) indicate that the location of the origin point and the destination point cannot be greater than the link length, with $L_o$ and $L_d$ are nonnegative continuous variables. To prevent the OD path from becoming excessively short, Eq (7) is utilized to ensure that the OD path includes at least one complete link. The number of complete links is one less than the number of endpoints in the OD path. Eq (8) formulates the total length of the OD path. The shortest path of the chosen OD is obtained by Dijkstra's algorithm, which is commonly used to solve the shortest path problem in weighted graphs, where in this paper the weight is the number of RSUs in each road.

$$0 \leq L_o \leq L_r \tag{5}$$

$$0 \leq L_d \leq L_r \tag{6}$$

$$\sum_{i_1 \in I} \sum_{i_2 \in I} s_{i_1 i_2}^{(o,d)} > 1 \tag{7}$$

$$L_s = L_o + L_d + L_r * \sum_{i_1 \in I} \sum_{i_2 \in I} s_{i_1 i_2}^{(o,d)} \tag{8}$$

## 3.4 Information transmission model

The most common mode of information communication in IoV is the online transmission. However, when there is no other CV within the communication range, online transmission

will be disrupted. In that case, the information moves forward with the vehicle, which is called offline transmission. Obviously, the distance that information moves by online transmission and offline transmission are alternating. We modify Zhang's time headway distribution model [29] by adding CV penetration in the data process stage and transferring it into a headway distribution model. Eq (9) calculates the headway distribution of the dataset with the given headway condition. Unlike some other headway distributions that are based on specific assumptions, this nonparametric distribution model with a Gaussian kernel function can be adapted to different flow conditions without any assumptions. Eq (10) is Zhang's time headway distribution model. Eq (11) is the best value of $h$ for this model.

$$f(x) = \frac{1}{m*h} \sum_{j}^{m} K\left(\frac{x - X_{\theta i}}{h}\right) * \bar{v} \tag{9}$$

$$f(x) = \frac{1}{m*h} \sum_{j}^{m} K\left(\frac{x - X_i}{h}\right) \tag{10}$$

$$h = \left(\frac{4\hat{\sigma}^5}{3m}\right)^{1/5} \tag{11}$$

We modify the information transmission model of Du and Dao in [17], the headway distribution model in it comes from Markov stochastic process, and the calculated offline transmission distance becomes infinite occasionally. The probability density function for the distance of the vehicle where the information is located and its nearest CV is used to calculate the distance in a single jump. The expectation of the number of hops in a single round of online transmission is addressed by mathematical induction. Combined with the distance and the expectation of the number of jumps in a single hop, Eq (12) represents the distance expectation of information advance in single-round online transmission.

$$E_{online} = \bar{k} \times \frac{\int_0^r sf(s)\Delta s}{\int_0^r f(u)\Delta u} \tag{12}$$

When the online transmission is interrupted, the information will transmit by offline transmission until another vehicle is connected. Similar to online transmission, the probability and distance expectation of offline transmission are computable. The distance of the information transmitted by the offline mode is determined by the velocity difference between the two vehicles. However, when their velocities are close or they have the same velocity, the offline transmission time goes to infinity. In that case, we impose a restriction on the transmission time, which is assumed to reselect a CV at time $\tau_0$. The modified equation and Du's equation in [17] are described by Eqs (13) and (14), respectively.

$$E_{offline} = v_1 \times \min\left(\frac{\frac{\int_r^{+\infty} sf(s)\Delta s}{1 - \int_0^r f(u)\Delta u} - r}{v_{12}}, \tau_0\right) \tag{13}$$

$$E_{offline} = v_1 \times \frac{\frac{\int_r^{+\infty} sf(s)ds}{1 - \int_0^r f(u)du} - r}{v_{12}} \tag{14}$$

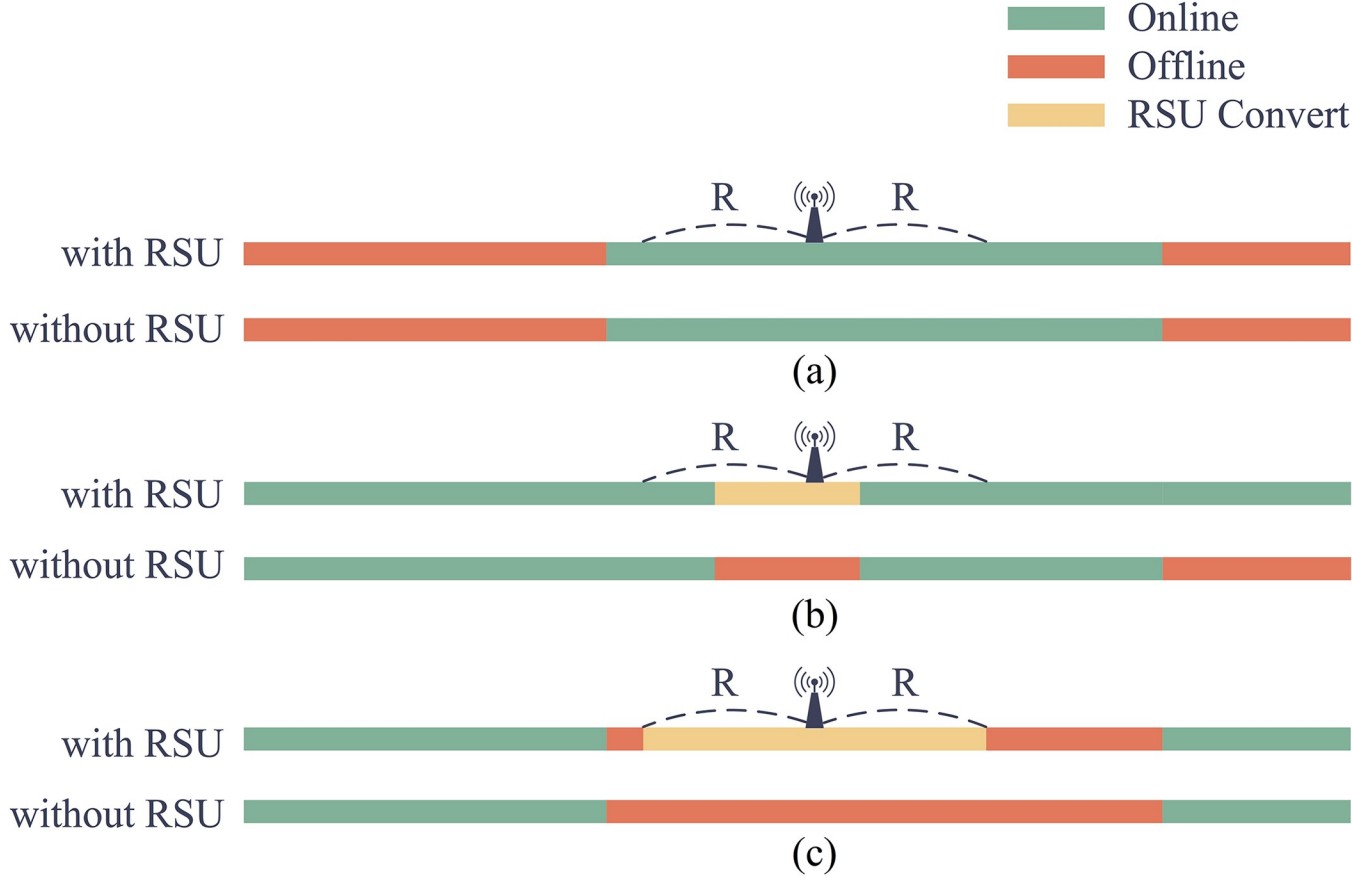

**Fig 2. The transition of transmission distance with RSU.** Corresponding to the three conditions in Eq (16). Different colors represent the distance information transmits through different modes.

### 3.5 Online transmission ratio

In the information transmission path, if the proportion of online transmission distance is below an acceptable threshold, the corresponding penalty cost will be added to the objective function (2) according to Eq (15).

$$p_\omega(\varphi) = \begin{cases} 1, & \dfrac{D_o(\varphi)}{D_t(\varphi)} \le \gamma \\[3mm] 0, & \dfrac{D_f(\varphi)}{D_o(\varphi)} > \gamma \end{cases} \tag{15}$$

As shown in Fig 2, the function of the RSU is to convert the offline transmission distance to the online transmission distance within its communication range.

Depending on the distribution of information transmission distances, the transformation effect of RSU can be classified into three conditions:

a. the RSU communication range is full of online transmission distances, and no conversion occurs.

b. the RSU communication range is full of offline transmission distances, converting 2R offline transmission distance into online transmission distance.

c. the RSU communication range contains part of the distance transmitted offline, it will be partially converted.

The convert function is formulated in Eq (16).

$$F_c = \begin{cases} 0, & condition(a) \\ D_c, & condition(b) \\ 2R, & condition(c) \end{cases} \tag{16}$$

Where $D_c$ denotes the distance converted from Offline to Online.

## 4. Solution algorithm

This section outlines the approach to improve the efficiency of the solution algorithm. We propose NNSGA-II based on NSGA-II. The core idea of NNSGA-II is as follows:

- Step 1: Randomly generate an initial population of 2N individuals and calculate the nondomination relationship and crowding distance.

- Step 2: Sort the population based on nondomination level and crowding distance and select the top half as the parent population.

- Step 3: Use binary tournament selection to randomly select two individuals from the parent population, calculate the crossover and mutation probabilities based on the average nondomination level of the selected individuals, and perform crossover and mutation operations. Repeat until generating an offspring population of the same size as the parent population.

- Step 4: Merge the parent and offspring populations, sort the population based on nondomination relationship and crowding distance, and select the top half as the parent population for the next generation. Repeat Step 2–4 until reaching the preset number of iterations or meeting the termination criteria.

NSGA-II is a multi-objective optimization algorithm that simulates the process of biological evolution. Since multi-objective optimization problems do not have a single optimal solution, but rather a Pareto Front of optimal solutions [30], traditional numerical algorithms that rely on gradient information and objective function constraints are not suitable. Instead, NSGA-II simulates biological mechanisms such as inheritance and mutation to randomly search for solutions and employs natural selection to preserve good solutions. NSGA-II is highly suitable for problems where it is difficult for traditional numerical algorithms to find Pareto solution sets in a reasonable time, such as the model proposed in this paper. NSGA-II achieves great results in bi-objective optimization and introduces a crowding degree to enrich population diversity. However, extensive experiments have demonstrated that NSGA-II has a low convergence efficiency, especially when the gene candidate values of individuals are not limited to binary.

In the proposed bi-objective optimization model, there are 6 candidate values for each gene of an individual. To improve the efficiency of the model solution, we modify the traditional NSGA-II algorithm by adjusting the probabilities of crossover and mutation based on the nondomination level of variables. Individuals with higher nondomination levels are assigned lower crossover and mutation probabilities to preserve better solutions, while individuals with lower nondomination levels are assigned higher crossover and mutation probabilities to better explore the search space and improve the global search ability of the algorithm. Therefore, we propose NNSGA-II, the steps of which are described as follows:

- Step 1: Population initialization. Generate a population of N individuals randomly, which corresponds to a chromosome, composed of $\alpha$ genes, each gene is encoded as an integer. The value of $\alpha$ is equal to the number of candidate links at RSU deployment in the network. A gene position of integer $\beta$ indicates that the number of RSUs on the corresponding link is $\beta$, and a gene position of 0 indicates that there are no RSUs on the corresponding link. In this model, $\alpha$ equal to 12 and the candidate values of $\beta$ are 0,1,2,3,4,5. Each chromosome represents one RSU deployment. Each population represents a set of RSU location options.

- Step 2: Fast Nondominated Sorting. Prior to the selection operation, the population is ranked according to the dominant and nondominant relationships between individuals: first, all nondominated individuals in the population are identified and assigned a shared virtual fitness value to obtain the first nondominated optimal layer. Then, ignoring the selected individuals, the rest of the individuals in the population are stratified according to dominant and nondominant relationships, giving a new virtual fitness value that is smaller than the value of the previous layer. Continue the above operation for the remaining individuals until all individuals in the population are stratified.

- Step 3: Calculate the crowding distance. The crowding distance is a measure of the distance between an individual and its neighbors. Greater average crowding distance leads to better population diversity. The individuals in the front queue are ranked according to the value of each objective function, and the crowding distance of the ranked individuals on the left side of the front queue is set as INFINITE. The crowding distance of each individual is then computed successively, and the crowding distance of the sorted individual to the right of the front of the queue is set to INFINITE.

- Step 4: Selection. Two individuals are randomly selected from the parent population, and the more adaptive individual is selected to enter the offspring population based on the nondomination level and crowding distance. In this way, two individuals are selected for crossover.

- Step 5: Crossover. The nondomination level for determining the crossover probability is the average of the two selected individuals' nondomination levels. Eq (17) shows the crossover probability is the product of the nondomination level and the fundamental crossover probability, and is used to determine whether a gene crossover has occurred. Genes are exchanged at a crossover point between two selected individuals, where the crossover point is set in the middle of the gene. After crossover, the two individuals are sent to Step 6.

$$p_{modify} = \frac{p_{default}}{2} * Level_{Nondomination} \tag{17}$$

- Step 6: Mutation. The mutation probability is the same as the crossover probability, which is used to determine whether there is a gene mutation or not. To perform a gene mutation, one traverses all genes and determines whether to perform a gene mutation, changing the value of the gene that needs to undergo a gene mutation. After mutation, the two individuals are added to the offspring population.

- Step 7: Generate new parent populations. Repeat steps 4 to 6 until the number of offspring reaches N, and then combine the offspring and parent populations. Perform steps 2 and 3 on the new population of 2N individuals and then sort them according to their nondomination level and crowding distance, keeping the top N individuals as the new parent population.

- Step 8: Check if the algorithm has exceeded the maximum number of iterations. If it has, terminate the algorithm, and output the current optimal solution and objective function value. If not, continue to Step 7.

Fig 3 illustrates the steps of the NNSGA-II algorithm integrated with the solution of the bi-objective optimization problem, highlighting the processes of initialization, non-dominated sorting, crossover, mutation, and the generation of new parent populations.

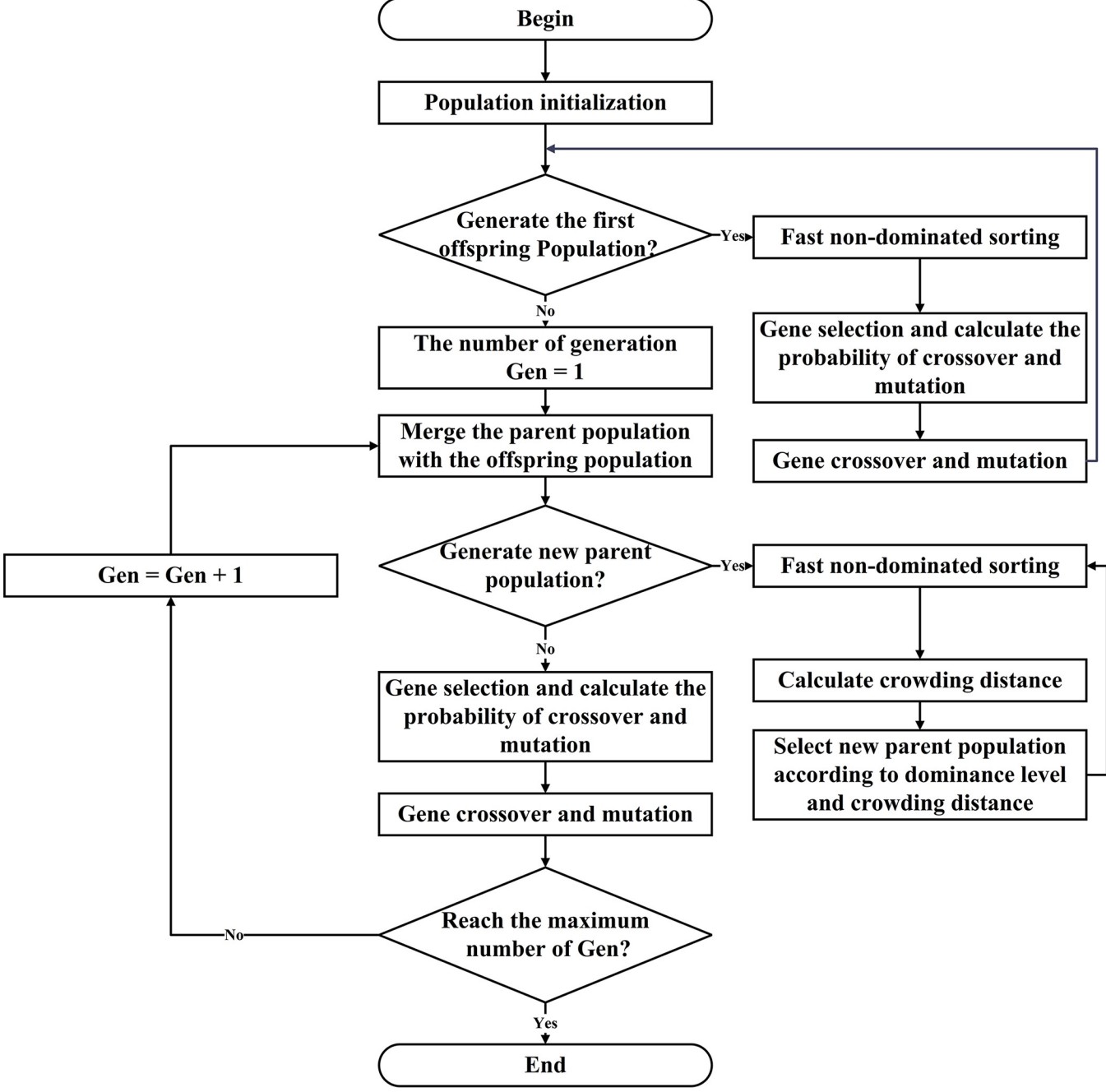

**Fig 3. The steps of the NNSGA-II integrated with the solution of bi-objective optimization problem.**

## 5. Numerical experiment

In this section, the proposed bi-objective optimization model and the NNSGA-II algorithm are evaluated through a series of numerical experiments. The first sub-section describes the experimental design. The second and third sub-sections evaluate the proposed model and NNSGA-II with other models and algorithms, respectively.

### 5.1 Experiment design

Experiments compare the bi-objective optimization model with the benchmark models under different traffic conditions and CV penetration. The effective measure is the amounts of RSUs and penalty cost associated with the online transmission distance ratio in the target model compared to the benchmark model. Different parameters are applied to analyze the sensitivity of the algorithm, including penetration, penalty threshold, population size, intersection influence, and penalty cost. The Pareto optimality and algorithm efficiency are used as effective measures to compare NNSGA-II with regular NSGA-II.

### 1) Compared Models Description

The compared models are described below:

- The benchmark model 1: the benchmark model 1 is also a bi-objective optimization model. Different from the proposed model, it does not take the transmission range of RSU into consideration. The candidate numbers of RSUs on each road are 0 and 1. The objective function and other constraints are the same as in the proposed model.

- The benchmark model 2: the benchmark model 2 is a deterministic deployment. The number of deployed RSUs on each road is the same, that is, six deterministic models are ranging from 0 to 5. Due to the fixed location and number of RSU, this model only computes the penalty cost for online transmission distance ratio below an acceptable threshold based on the changing traffic conditions.

- The benchmark model 3: the benchmark model 3, as introduced in Liang's seminal paper [19], is a significant addition to our comparative analysis. This model aims to minimize the sum of the cost associated with RSU investment and the expectation of the penalty cost associated with V2R communication delay exceeding a pre-determined threshold, has been chosen due to its relevance to our study objectives.

- The proposed model: more information can be found in Section 3.

### 2) Compared Algorithms Description

The compared algorithms described below:

- The benchmark algorithm 1: the benchmark algorithm 1 is Genetic Algorithm (GA) [31], to solve the benchmark model 2. GA provides robust and near-optimal solution for NP-hard problems in the literature [32,33].Simulating natural evolutionary processes to search for optimal solutions, GA transforms the problem-solving process into a process similar to the selection, crossover, and mutation of chromosome genes in biological evolution.

- The benchmark algorithm 2: The conventional NSGA-II proposed in [34] is used as the benchmark algorithm 2 to showcase the improved quality and efficiency of the proposed algorithm. The conventional NSGA-II has the same steps as NNSGA-II.

- NNSGA-II: Compared to conventional NSGA-II, NNSGA-II makes some changes in crossover and mutation. Details about this algorithm can be found in Section 4.

### 3) Parameters Setting

We have made a series of assumptions regarding the communication protocols and scenarios for V2V and V2R communications. Firstly, we assume the adoption of an efficient communication protocol, such as IEEE 802.11p, to ensure reliable data transmission and low latency. Additionally, we consider the communication range of each RSU, where vehicles can only communicate with RSUs within their respective coverage areas. The deployment density is adjustable to meet the communication requirements of different regions. We also consider the possibility of communication channels being affected by other vehicle communications or external interference sources. Furthermore, we assume that the data transmission rate in communication is sufficiently high to support real-time transmission requirements. Lastly, we assume that communication protocols and equipment can ensure data integrity and security to prevent data tampering or unauthorized access. These assumption conditions are crucial in our research, aiding in establishing models and analyzing the feasibility of communication.

The experiments are performed on a grid network with 12 nodes and 24 road sections, the length of each link is 10km and the structure of which is shown in Fig 1. In this experiment, each location on each link can be used as a starting point and destination for information transmission. For the purpose of avoiding too close the distance between the starting point and the destination point, Eq (7) is used to restrict the shortest path between OD to contain at least one complete road section. The communication range of both V2V and V2R is set to 1000 m [35]. The total traffic flow is planned according to 6 different Levels of Service (LOS) in HCM [36], and different vehicle situations are randomly generated by SUMO (Simulation of Urban Mobility) for each LOS. The default values of parameters are listed in Table 3. These defaults serve as an example to illustrate the advantages of the NNSGA-II over other algorithms. It can be found that the performance of the NNSGA-II is typically superior at all tested levels. A sensitivity analysis of parameters such as vehicle penetration rate, road service level, and other factors that may affect the experimental results is presented in the following two subsections. The proposed model appears to be insensitive to the specific choices of parameter values.

We utilize a grid network as our research environment and conduct model training on a Windows 11 operating system with an Intel i7-12700H 2.30 GHz processor and an RTX 3060 GPU equipped with 6 GB of memory. Using the Traci interface of SUMO, we are able to obtain real-time traffic data from the simulation environment. The duarouter tool provided by

**Table 3. The default values of parameters.**

| Parameter | Default Value |
| --- | --- |
| RSU deployment and maintenance cost | 1 |
| penalty cost | 1 |
| CV penetration | 0.6 |
| threshold of the proportion of online information transmission distance | 0.95 |
| LOS | A |
| basic crossover probability | 0.9 |
| basic mutation probability | 0.08 |
| influence of intersection | 0.9 |
| population of gene | 100 |

SUMO randomly generates routes based on the set simulation time and the total number of vehicles. The total number of vehicles is determined by multiplying the sum of vehicles traveling between all different OD pairs by a hyperparameter. We select 25 different OD pairs from the road network, and the number of vehicles traveling between these pairs is randomly generated with a mean of 400 and a standard deviation of 200. The final hyperparameter that determines the total number of vehicles in the network is set to 0.6.

In the SUMO simulation, we obtain information about the location, speed, and other properties of each vehicle at the link of interest. The simulation was set to run for 7200 seconds and we collected vehicle information for a duration of 3600 seconds, ranging from 1800 seconds to 5400 seconds. This collected vehicle information serves as the input to the IoV information propagation model. This allows for the computation of the path of information propagation, as well as the distribution of online and offline transmission distances during the propagation process, given a randomly selected information origin-destination pair. Our bi-objective optimization model is able to analyze a large amount of distance distribution information to obtain a pareto set of relatively optimal RSU deployments.

## 5.2 Results of model evaluation

Under default traffic conditions, our model achieves notable advantages. When comparing models, to further understand the impact of different LoS on the experimental results, various scenarios from the optimal LoS A to the worst LoS F are considered. The deployment of RSUs is uniformly distributed along each road according to the number of RSUs. As benchmark model 1 does not consider the transmission distance of the RSUs, its allocation scheme allows only one RSU to be located at the center of each road. Fig 4A compares the Pareto solution sets of Benchmark Model 1 and the proposed model. The LoS of the proposed model is A, while the other parameters are the same as in Benchmark Model 1. It can be observed that when the LoS is above F, i.e., the level of extreme traffic congestion has not been reached, the Pareto solution set obtained by Benchmark Model 1 is a single point [0, 60]. On the other hand, with LoS set to F, the IoV transmission efficiency can be improved with only a modest number of RSUs. These two phenomena arising from different LoS can be understood as two distinct cases. The former is an RSU-driven IoV because the number of vehicles is insufficient to support its normal operation. The latter can be viewed as a vehicle-driven IoV because the number of vehicles is sufficient to support its normal operation and the RSUs are only complementary. Since the focus of this study is on RSU location optimization, the effect of sudden changes in IoV efficiency due to changes in the number of vehicles is not analyzed.

It should be noted that the change in IoV efficiency caused by the penetration rate of the CV is similar to the change caused by the LoS. The performance of Benchmark Model 1 is comparable to the proposed model when CVs have a high penetration rate, indicating that RSUs have a limited effect on information propagation stability under such conditions.

The number of RSUs in Benchmark model 2 assigned to each road is the same. Fig 4B shows a comparison between the Pareto set obtained from benchmark model 2 and the proposed model. Since benchmark model 2 fails to generate reasonable RSU deployment solutions under default conditions, the results of benchmark model 2 for different road service levels are added. The Pareto set of the proposed model is obtained under the default condition. It can be observed that in benchmark model 2, increasing the number of RSUs before covering all road sections does not improve the performance of the IoV when the LoS is above level D. In contrast, the proposed model can achieve better results at level A than those obtained from benchmark model 2 at level E. Only when the LoS reaches level F can benchmark model 2 achieve

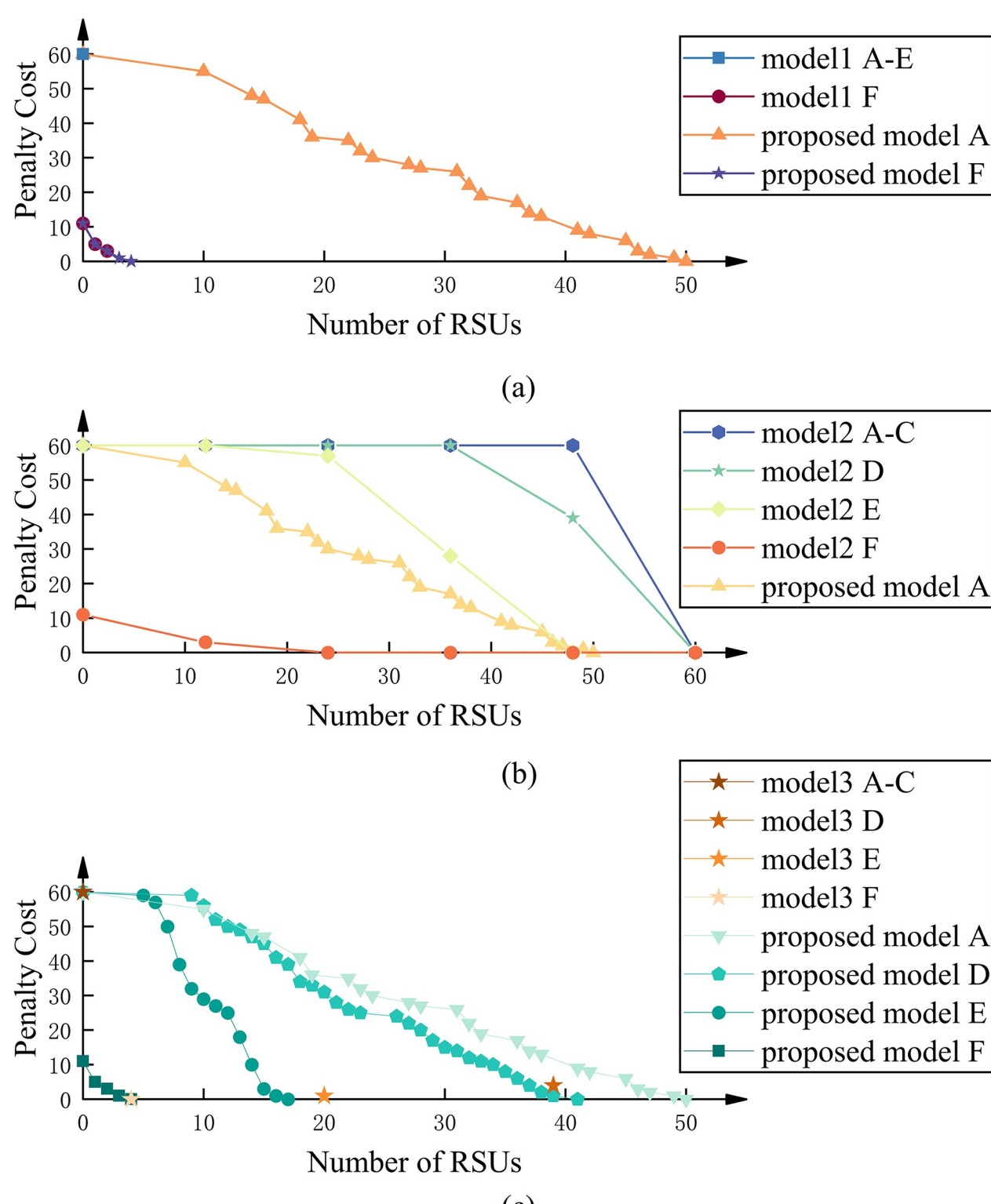

**Fig 4. Results of model evaluation.** A to F represent various LoS as defined in HCM, while different model corresponds to Section 5.1, subsection 1).

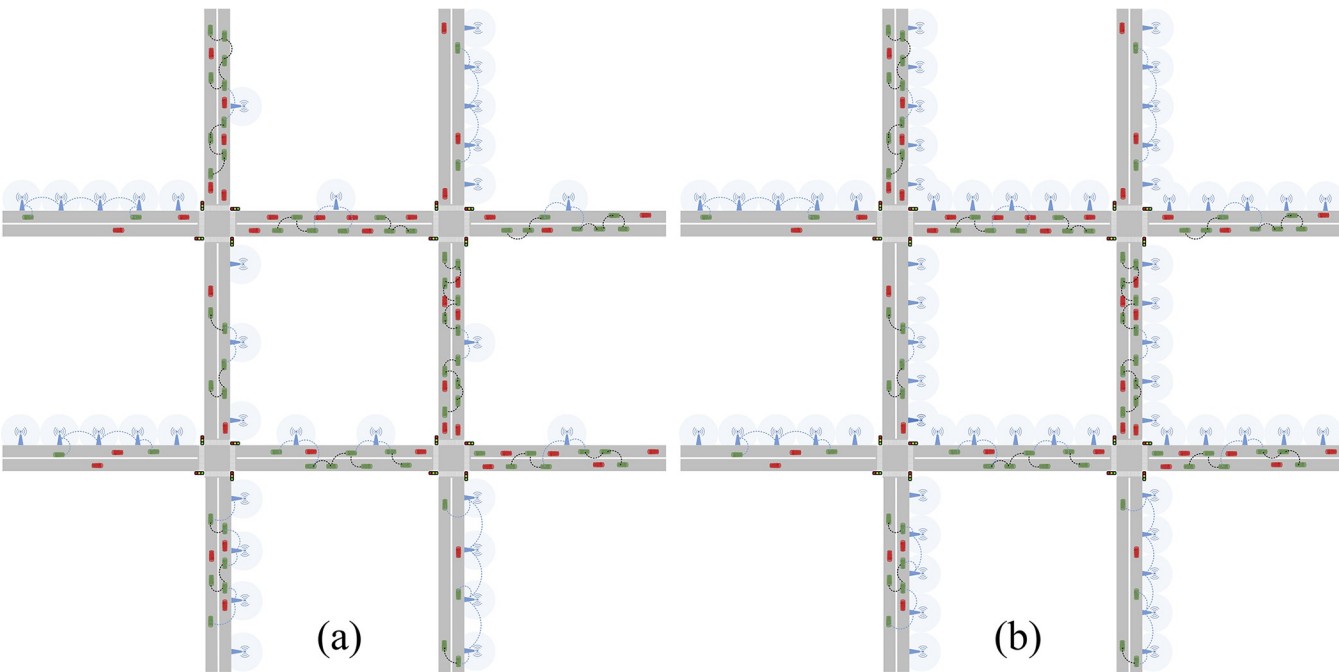

**Fig 5. Comparison of final RSU deployment between proposed and baseline models under default conditions.**

relatively ideal vehicular network performance, although the impact of RSUs at this condition is minimal.

Benchmark model 3 aims to minimize the sum of penalty cost and the cost associated with RSU. As shown in Fig 4C, benchmark model 3 is better than benchmark models 1 and 2, it can even get the same result as a Pareto solution set of the proposed model. However, when the LoS is above F, the results of the proposed model are significantly better than those of benchmark model 3. Similar to the previous findings, RSUs have few helps on the IoV efficiency at the level of F. Therefore, it can be concluded that the proposed model has a significant advantage over the three benchmark models in a RUS-dominated IoV.

Fig 5 demonstrates the final RSU distribution results of the proposed model compared to three baseline models under default conditions. Notably, the final RSU distribution results of the three baseline models are identical, utilizing 60 RSUs to achieve full coverage of the road segment, as shown in Fig 5A. In contrast, the proposed model accomplishes the same outcome using only 33 RSUs, as shown in Fig 5B. The comparison highlights that the optimized model significantly differs from traditional uniform distribution strategies in terms of RSU deployment.

Our model demonstrates strong adaptability to different types of road networks. Whether in urban areas with dense and complex road networks or in suburban areas with simpler and more linear road structures, the model can effectively optimize RSU deployment. This adaptability is crucial because it ensures that the model can be applied to a wide range of real-world scenarios, enhancing its practical value. For urban areas, where traffic congestion and high vehicle density are common, the model's ability to dynamically adjust RSU placement based on traffic flow patterns is particularly beneficial. By deploying more RSUs in less congested lanes, the model ensures that communication coverage is maximized, even in areas with high traffic volumes. This approach not only improves the reliability of the IoV but also optimizes resource utilization. In suburban areas, where traffic flow is generally more uniform and

predictable, the model's flexibility allows for a more efficient distribution of RSUs. By placing RSUs strategically to cover longer stretches of road, the model minimizes the total cost while maintaining high transmission efficiency. This is especially useful in scenarios where the budget for RSU deployment is limited.

### 5.3 Results of solution algorithm evaluation

Regarding the proposed model, both the NSGA-II and NNSGA-II algorithms yield the same Pareto set, nevertheless, NNSGA-II shows a faster convergence speed. This section compares the iterative procedure of the two algorithms under different parameters.

The two algorithms are compared by Hypervolume (HV) metric [37], which is commonly used to evaluate the quality of Pareto sets. In addition, we propose the penalty cost value decreases with each additional RSU (PDWR), which evaluates the penalty cost reduction associated with adding an RSU. Physically, models with higher values of this metric can better improve the stability of the IoV. Fig 6 shows the iterative results for the two metrics in the Pareto set obtained from the training of the two algorithms. It should be noted that, for ease of observation, the curves are not started at iteration 0 and the results are averaged over ten runs.

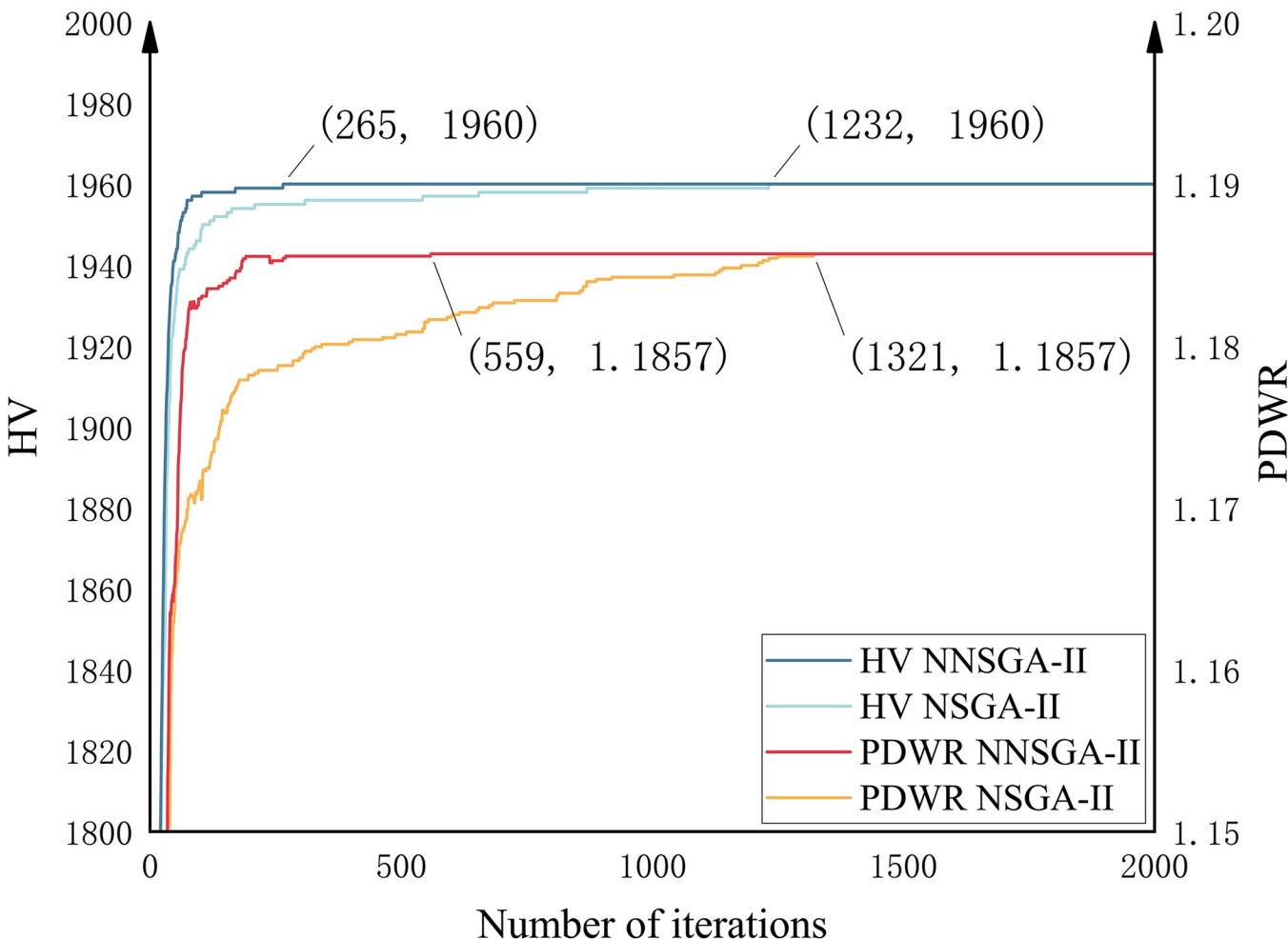

**Fig 6. Comparison of HV and PDWR convergence results between NSGA-II and NNSGA-II.** The coordinates of the four marked points represent the locations where the algorithm converges.

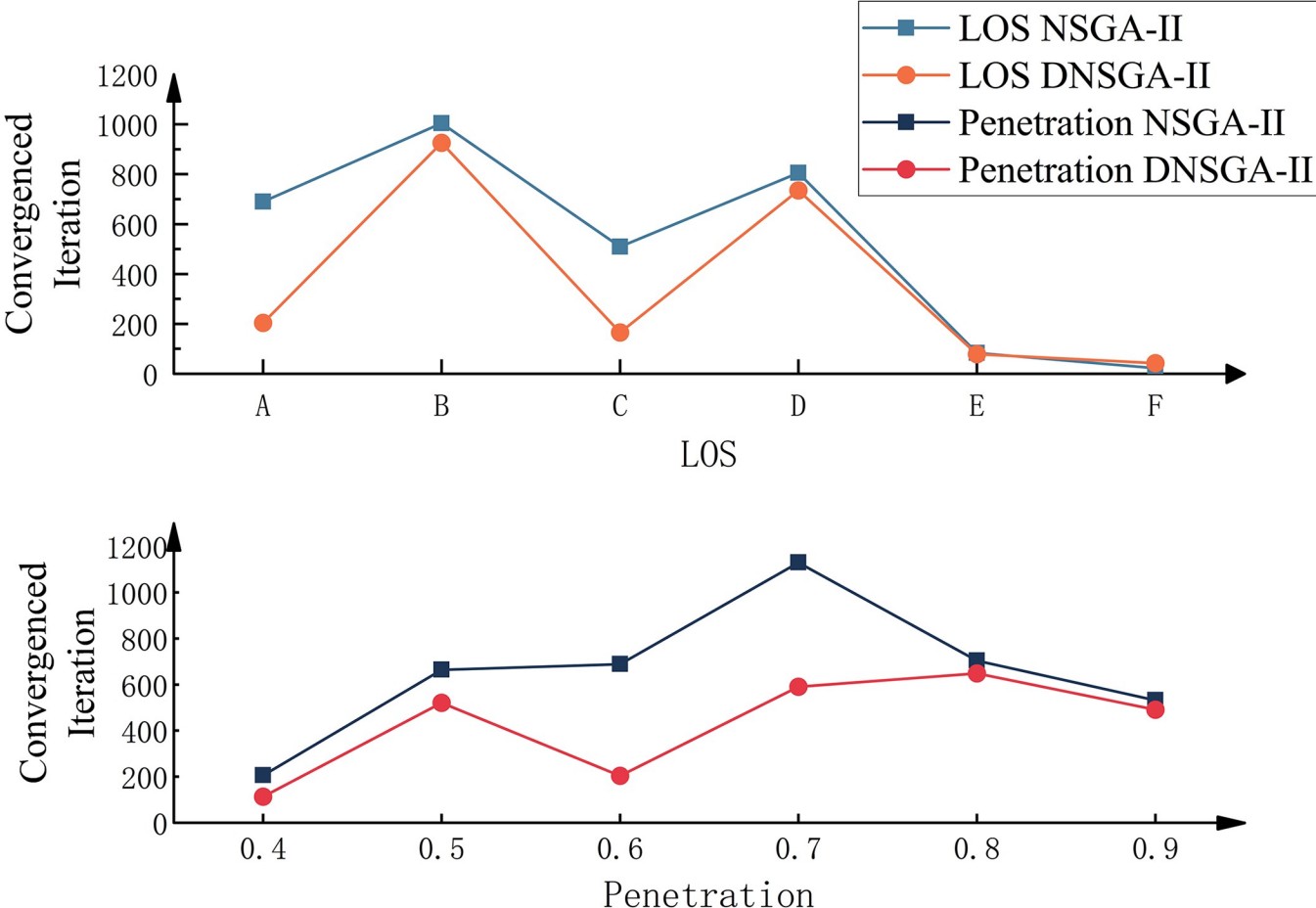

**Fig 7. Comparison of convergence results between NSGA-II and NNSGA-II under different (LoS) and penetration values.**

The NNSGA-II algorithm achieves convergence of the HV and PDWR metrics at the 265th and 559th iterations, while the NSGA-II algorithm achieves convergence of the HV and PDWR metrics at the 1232nd and 1321st iterations. Compared to the NSGA-II algorithm, the NNSGA-II algorithm has a significant advantage in terms of training speed.

Experiments show that NNSGA-II converges significantly faster than NSGA-II under different LoS and penetration rates. Fig 7 compares the convergence iterations of the two algorithms for different LoS and penetration rates. Similar to Fig 6, the average of ten executions is reported in the results. It can be seen from the figure that only when the LoS is F, the convergence speed of NSGA-II is slightly better than that of NNSGA-II. Under other conditions, NNSGA-II has a more pronounced advantage in terms of convergence speed.

Fig 8 shows the test results of the convergence speed between NSGA-II and NNSGA-II, which is obtained by averaging the results of ten runs. These results demonstrate the high robustness of NNSGA-II in the RSU deployment problem.

Robustness analysis is performed to compare the impact of different penalty costs, coverage thresholds, the impact of intersections, and population size on the performance of NNSGA-II and NSGA-II algorithms. The impact of intersections on V2V transmission is complex [38], for ease of understanding, we use several constants to represent. The results show that NNSGA-II outperforms NSGA-II in different scenarios. NNSGA-II achieves a higher

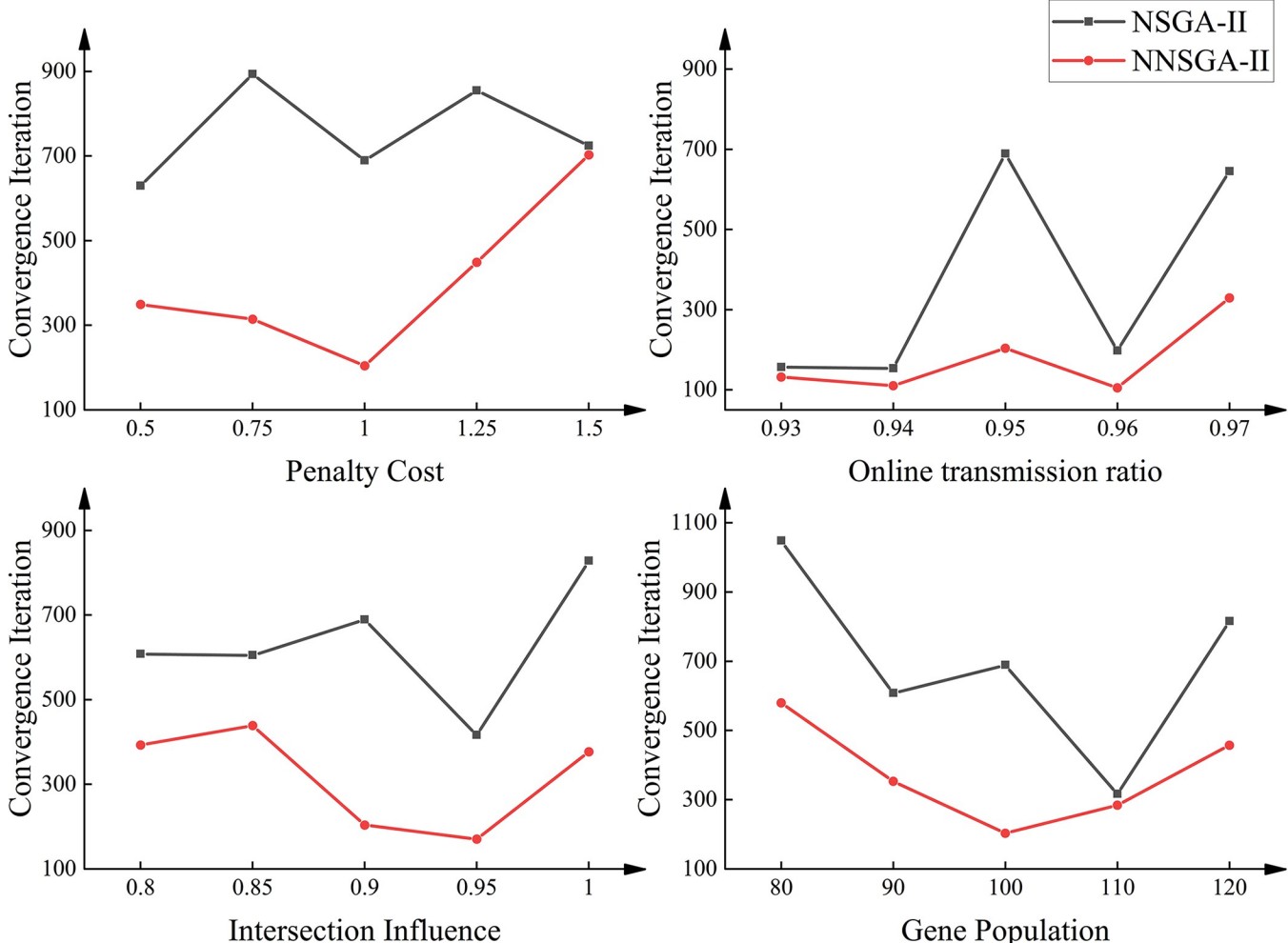

**Fig 8. Results of the convergence speed between NSGA-II and NNSGA-II on different penalty costs, online transmission ratios, intersection influences, gene populations.**

convergence level in a shorter time under different penalty costs. NNSGA-II also yields better results in different coverage ranges and at intersections. Moreover, NNSGA-II is able to achieve superior results even with smaller population size.

The robustness test results of the crossover and mutation probabilities for NSGA-II and NNSGA-II are shown in Fig 9. A total of 25 different combinations of crossover and mutation probabilities are selected for the experiment. To enhance readability, the data is smoothed by calculating the mean values of neighboring points. The two 3D surfaces do not intersect, which means that under all crossover and mutation probability conditions in the experiment, NNSGA-II requires fewer iterations to converge than NSGA-II.

## 6. Conclusion

We investigate the optimization of RSU deployment based on the IoV information transmission model under different LoS and IoV penetration in this paper. It provides a solution to solve the bi-objective optimization problem of IoV reliability and RSU cost, which can find the lowest cost RSU deployment scheme under the premise of ensuring IoV reliability. The IoV reliability is represented by the cumulative number that the proportion of online transmission

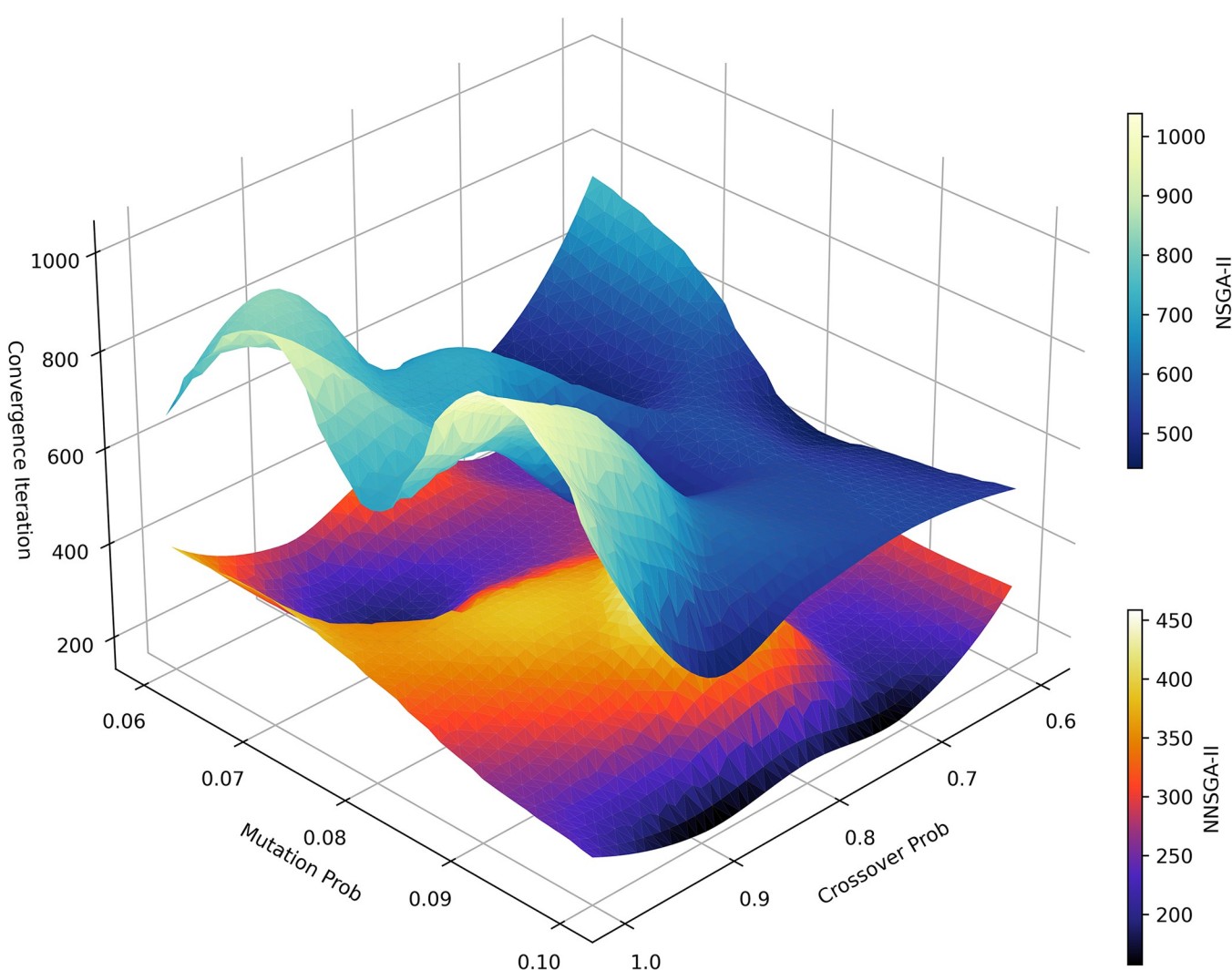

**Fig 9. Robustness test results of the crossover and mutation probabilities for NSGA-II and NNSGA-II.**

distance below a threshold, which is used as one of the objectives in a bi-objective optimization model along with the cost of RSUs. In order to improve the efficiency of the model's solution, we make modifications to the traditional NSGA-II by adjusting the crossover and mutation probabilities based on the nondomination level. In addition, the model and algorithm are compared with three benchmark models and two different algorithms, respectively. The experimental results can be concluded as follow:

- The proposed model has a great advantage compared with other benchmark models in RSU-dominated IoV, while having comparable performance in CV-dominated IoV.

- The distance-based reliability proposed in this paper demonstrates superior effectiveness and adaptability in the RSU deployment optimization problem compared to the time-based reliability, which aims to reduce the total information transmission time.

- The solution sets obtained by NSGA-II and NNSGA-II are the same under different conditions. NNSGA-II is 10.01% more computationally efficient than the traditional genetic algorithm.

While our model demonstrates robust performance and adaptability across different road network types and traffic conditions, there are areas for improvement, particularly in handling dynamic traffic patterns and reducing computational complexity. These insights will guide future research and development efforts to enhance the model's practical applicability. Future research can be conducted from two perspectives: one is to propose new IoV research models based on distance reliability, and the other is to continue improving the efficiency of algorithms to adapt to larger-scale road networks.

## Author Contributions

**Conceptualization:** Jun Zhang, Guangtong Hu.

**Data curation:** Guangtong Hu.

**Formal analysis:** Guangtong Hu.

**Funding acquisition:** Jun Zhang.

**Investigation:** Guangtong Hu.

**Methodology:** Guangtong Hu.

**Project administration:** Jun Zhang.

**Resources:** Guangtong Hu.

**Software:** Jun Zhang.

**Supervision:** Guangtong Hu.

**Validation:** Guangtong Hu.

**Visualization:** Guangtong Hu.

**Writing – original draft:** Guangtong Hu.

**Writing – review & editing:** Jun Zhang.

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
