## [Decision Letter · Decision Letter 0]

22 Oct 2024

PONE-D-24-39904Road side unit deployment optimization for the reliability of Internet of Vehicles based on information transmission modelPLOS ONE

Dear Dr. Hu,

Thank you for submitting your manuscript to PLOS ONE. After careful consideration, we feel that it has merit but does not fully meet PLOS ONE’s publication criteria as it currently stands. Therefore, we invite you to submit a revised version of the manuscript that addresses the points raised during the review process.

ACADEMIC EDITOR: Major revision

We look forward to receiving your revised manuscript.

Kind regards,

Agbotiname Lucky Imoize

Academic Editor

PLOS ONE

Journal Requirements:

1. When submitting your revision, we need you to address these additional requirements. Please ensure that your manuscript meets PLOS ONE's style requirements, including those for file naming. The PLOS ONE style templates can be found at https://journals.plos.org/plosone/s/file?id=wjVg/PLOSOne_formatting_sample_main_body.pdf and https://journals.plos.org/plosone/s/file?id=ba62/PLOSOne_formatting_sample_title_authors_affiliations.pdf 2. Please note that PLOS ONE has specific guidelines on code sharing for submissions in which author-generated code underpins the findings in the manuscript. In these cases, we expect all author-generated code to be made available without restrictions upon publication of the work. Please review our guidelines at https://journals.plos.org/plosone/s/materials-and-software-sharing#loc-sharing-code and ensure that your code is shared in a way that follows best practice and facilitates reproducibility and reuse. 3. Thank you for stating the following financial disclosure: "This study is supported by grants from the National Social Science Foundation of China (grant number 20BGL001), which is required to be unique." Please state what role the funders took in the study.  If the funders had no role, please state: "The funders had no role in study design, data collection and analysis, decision to publish, or preparation of the manuscript." If this statement is not correct you must amend it as needed. Please include this amended Role of Funder statement in your cover letter; we will change the online submission form on your behalf. 4. Thank you for stating the following in the Acknowledgments Section of your manuscript: "This study is supported by grants from the National Social Science Foundation of China (grant number 20BGL001), which is required to be unique." We note that you have provided funding information that is not currently declared in your Funding Statement. However, funding information should not appear in the Acknowledgments section or other areas of your manuscript. We will only publish funding information present in the Funding Statement section of the online submission form. Please remove any funding-related text from the manuscript and let us know how you would like to update your Funding Statement. Currently, your Funding Statement reads as follows: "This study is supported by grants from the National Social Science Foundation of China (grant number 20BGL001), which is required to be unique." Please include your amended statements within your cover letter; we will change the online submission form on your behalf.

Additional Editor Comments:

The authors should revise the paper based on the reviewers comments.

Reviewers' comments:

Reviewer's Responses to Questions

**Comments to the Author**

1. Is the manuscript technically sound, and do the data support the conclusions?

Reviewer #1: Partly

Reviewer #2: Yes

2. Has the statistical analysis been performed appropriately and rigorously? 

Reviewer #1: No

Reviewer #2: Yes

3. Have the authors made all data underlying the findings in their manuscript fully available?

Reviewer #1: Yes

Reviewer #2: No

4. Is the manuscript presented in an intelligible fashion and written in standard English?

Reviewer #1: Yes

Reviewer #2: Yes

5. Review Comments to the Author

Reviewer #1: The presentation of the NNSGA-II algorithm focuses on optimizing RSU (Road Side Unit) deployment within the Internet of Vehicles (IoV) information transmission model. The exploration of various Levels of Service (LoS) reveals their effects on IoV reliability and RSU costs, culminating in a proposed bi-objective optimization framework. Here are my concerns and suggestions to the authors:

1. The presence of formatting issues with some equations detracts from the clarity of mathematical reasoning.

2. The description of the NNSGA-II algorithm could be enhanced by adding flowcharts or pseudocode for better understanding.

3. A need exists for more detailed examples and visualizations to illustrate the algorithm's practical applications, with current examples lacking robustness.

4. The model's adaptability to different road network types and limitations observed during simulations requires clarification.

5. The rationale behind selecting specific parameter values for mutation and crossover probabilities needs further explanation, alongside any sensitivity analysis conducted.

6. This paper develops a bi-objective optimization model utilizing the NNSGA-II algorithm for improved RSU deployment strategies in IoV systems. The effective comparison of the proposed model with benchmark methods highlights notable advantages in specific scenarios. The need for improvements in mathematical clarity, inclusion of visual aids, and detailed examples is crucial. Insights into parameter selection and model adaptability to various road types are encouraged. With these enhancements, the work is positioned to yield significant insights into practical applications in smart transportation systems.

7. The necessity exists for a detailed simulation comparison to elucidate the proposed model's performance. Further exploration of the model's algorithms is essential for bolstering claims of efficiency.

8. Expanding evaluation breadth to encompass various scenarios enhances robustness.

9. Increasing figure size is crucial for improved interpretation of key data and trends.

10. The adaptation of the model to distinct environmental scenarios warrants exploration.

11. Conducting a thorough cost-benefit analysis is critical for understanding economic impacts. An additional explanations or examples of the applied NSGA-II algorithm could help non-expert readers understand its implementation and significance.

12. While numerical results are mentioned, consider providing more details about the test scenarios, conditions, and parameters used in the simulations to enhance reproducibility and understanding of the results.

Reviewer #2: This paper explores optimizing Road-Side Unit (RSU) deployment for Internet of Vehicles (IoV) based on different Line of Sight (LoS) conditions and IoV penetration rates. It aims to minimize the cost of RSU deployment while ensuring IoV reliability through a bi-objective optimization model. The reliability is measured by the proportion of online transmission distance below a certain threshold. The modified NSGA-II algorithm, with adjustments to crossover and mutation probabilities, is used to improve solution efficiency. Experimental results show the proposed model performs better in RSU-dominated scenarios and that NNSGA-II is more efficient than traditional algorithms by 10.01%.

1. Introduction should cover the background of the area, problem motivation, importance of problem, methodology used, results analysis and concluding remarks in a short note.

2. The authors should include a section for literature survey (Section II- Related work).

3. Authors should very clearly state how does your work advances the state of the art. At the end of Section II, the authors should give a brief summary of related work to show the difference and improvement of this work. Maybe including a table summarizing the related work could help identifying gaps that your work fills.

4. I suggest to keep a table at the end of section II.

5. Section II may be divided into subsections like conventional methods, cooperative methods, non-cooperative methods, online/offline methods and their merits and demerits.

6. The simplicity of Figure 1 is noted, and it is suggested that more details be incorporated to depict the communication between CVs, and RSUs. This addition would provide greater clarity to the reader regarding the entire system model.

7. This submission lacks a comprehensive discussion of pertinent works, notably omitting references to key sources such as

a. "Contact duration-aware cooperative cache placement using genetic algorithm for mobile edge networks." Computer Networks 193 (2021): 108062.

b. Service Caching and User Association in Cache Enabled Multi-UAV Assisted MEN for Latency-Sensitive Applications.” Computers and Electrical Engineering (Elsevier) (2024)

6. PLOS authors have the option to publish the peer review history of their article (what does this mean?). If published, this will include your full peer review and any attached files.

Reviewer #1: **Yes: **Ebrahim E. Elsayed

Reviewer #2: No

---

## [Author Response · Author response to Decision Letter 0]

5 Nov 2024

Capital University of Economic and Business

Beijing, China

charles3000@cueb.edu.cn

November 2, 2024

Dear Editors

We thank the reviewers for their generous comments on the manuscript and have edited the manuscript to address their concerns.

We add a section about Related work, focusing on the contributions of related literature and improvements of our model. Additionally, we provide more detailed descriptions of the model content and include figures of results. Furthermore, we add the description of the model's performance in different traffic scenarios.

We believe that the manuscript is now suitable for publication in PLOS One.

Dr. Guangtong Hu

Ph.D. Candidate in Capital University of Economic and Business

On behalf of all authors. 

Reviewer 1 (Ebrahim E. Elsayed)

1. The presence of formatting issues with some equations detracts from the clarity of mathematical reasoning.

Answer: Thank you for your reminder. We check all the formulas and revise the format of Equations (15) and (16).

2. The description of the NNSGA-II algorithm could be enhanced by adding flowcharts or pseudocode for better understanding.

Answer: We add a description of the flowchart for the NNSGA-II algorithm in Figure 3.

3. A need exists for more detailed examples and visualizations to illustrate the algorithm's practical applications, with current examples lacking robustness.

Answer: We add visual results (Fig 5) and detailed descriptions to illustrate the practical application of the model and algorithm.

4. The model's adaptability to different road network types and limitations observed during simulations requires clarification.

Answer: We add the model's adaptability to different types of road networks in Section 5.2 and discuss the limitations observed during the simulation process in the Conclusion.

5. The rationale behind selecting specific parameter values for mutation and crossover probabilities needs further explanation, alongside any sensitivity analysis conducted.

Answer: In our study, the selection of specific mutation and crossover probability parameters is based on extensive experimental data. We conduct a wide-ranging sensitivity analysis to verify the model's sensitivity to different parameter values. Figure 9 illustrates the sensitivity analysis of mutation and crossover probabilities.

6. This paper develops a bi-objective optimization model utilizing the NNSGA-II algorithm for improved RSU deployment strategies in IoV systems. The effective comparison of the proposed model with benchmark methods highlights notable advantages in specific scenarios. The need for improvements in mathematical clarity, inclusion of visual aids, and detailed examples is crucial. Insights into parameter selection and model adaptability to various road types are encouraged. With these enhancements, the work is positioned to yield significant insights into practical applications in smart transportation systems.

Answer: We have addressed the reviewer's comments by improving mathematical clarity, adding visual aids and detailed examples, and providing insights into parameter selection and model adaptability to various road types. These enhancements significantly strengthen the practical applicability and robustness of our work.

7. The necessity exists for a detailed simulation comparison to elucidate the proposed model's performance. Further exploration of the model's algorithms is essential for bolstering claims of efficiency.

Answer: We demonstrate the performance of the proposed model in different traffic scenarios compared to other models in Section 5.2, showcase the superior performance of the proposed algorithm through comparisons in Section 5.3, and present the results of sensitivity analysis for various parameters in Section 5.4.

8. Expanding evaluation breadth to encompass various scenarios enhances robustness.

Answer: We compare six different traffic states, ranging from smooth to extremely congested, in Section 5.2.

9. Increasing figure size is crucial for improved interpretation of key data and trends.

Answer: Thank you for your reminder. We increase the size of all figures.

10. The adaptation of the model to distinct environmental scenarios warrants exploration.

Answer: We compare six different traffic states, ranging from smooth to extremely congested, in Section 5.2.

11. Conducting a thorough cost-benefit analysis is critical for understanding economic impacts. An additional explanations or examples of the applied NSGA-II algorithm could help non-expert readers understand its implementation and significance.

Answer: We add a discussion on the cost-benefit analysis and socioeconomic impacts of RSU deployment optimization in the introduction section. In Section 4, we provide additional descriptions and explanations of the NNSGA-II algorithm.

12. While numerical results are mentioned, consider providing more details about the test scenarios, conditions, and parameters used in the simulations to enhance reproducibility and understanding of the results.

Answer: We add details about the test scenarios, conditions, and parameters used in the simulations in Section 5.1.

Reviewer 2 (Anonymous)

1. Introduction should cover the background of the area, problem motivation, importance of problem, methodology used, results analysis and concluding remarks in a short note.

Answer: We add the results analysis and conclusions in the introduction section.

2. The authors should include a section for literature survey (Section II- Related work).

Answer: Thank you for your reminder. We add the Section II - Related work.

3. Authors should very clearly state how does your work advances the state of the art. At the end of Section II, the authors should give a brief summary of related work to show the difference and improvement of this work. Maybe including a table summarizing the related work could help identifying gaps that your work fills.

Answer: We summarize the related work and describe the improvements of our work at the end of Section 2. Additionally, we add a table to highlight the limitations of related work and our solutions.

4. I suggest to keep a table at the end of section II.

Answer: In Section 2, we add a table to highlight the limitations of related work and our solutions.

5. Section II may be divided into subsections like conventional methods, cooperative methods, non-cooperative methods, online/offline methods and their merits and demerits.

Answer: We categorize the related literature into four different types: Integer Programming, Heuristic Algorithms, Mathematical Modeling and Optimization Algorithms, Hybrid Management Strategies.

6. The simplicity of Figure 1 is noted, and it is suggested that more details be incorporated to depict the communication between CVs, and RSUs. This addition would provide greater clarity to the reader regarding the entire system model.

Answer: Thank you for your reminder. We re-draw Figure 1, and adding more details.

7. This submission lacks a comprehensive discussion of pertinent works, notably omitting references to key sources such as: a. "Contact duration-aware cooperative cache placement using genetic algorithm for mobile edge networks." Computer Networks 193 (2021): 108062. b. Service Caching and User Association in Cache Enabled Multi-UAV Assisted MEN for Latency-Sensitive Applications.” Computers and Electrical Engineering (Elsevier) (2024)

Answer: We conduct a thorough review of the literature and include citations in Section 5.

---

## [Decision Letter · Decision Letter 1]

1 Dec 2024

Road side unit deployment optimization for the reliability of Internet of Vehicles based on information transmission model

PONE-D-24-39904R1

Dear Dr. Hu,

We’re pleased to inform you that your manuscript has been judged scientifically suitable for publication and will be formally accepted for publication once it meets all outstanding technical requirements.

Kind regards,

Agbotiname Lucky Imoize

Academic Editor

PLOS ONE

Additional Editor Comments (optional):

The authors have revised the paper accordingly.

Reviewers' comments:

Reviewer's Responses to Questions

**Comments to the Author**

1. If the authors have adequately addressed your comments raised in a previous round of review and you feel that this manuscript is now acceptable for publication, you may indicate that here to bypass the “Comments to the Author” section, enter your conflict of interest statement in the “Confidential to Editor” section, and submit your "Accept" recommendation.

Reviewer #1: (No Response)

2. Is the manuscript technically sound, and do the data support the conclusions?

Reviewer #1: (No Response)

3. Has the statistical analysis been performed appropriately and rigorously? 

Reviewer #1: (No Response)

4. Have the authors made all data underlying the findings in their manuscript fully available?

Reviewer #1: (No Response)

5. Is the manuscript presented in an intelligible fashion and written in standard English?

Reviewer #1: (No Response)

6. Review Comments to the Author

Reviewer #1: The paper has been suitably revised. I recommend acceptance.

I am happy with the responses and edits of the manuscript, and they address all of my concerns.

7. PLOS authors have the option to publish the peer review history of their article (what does this mean?). If published, this will include your full peer review and any attached files.

Reviewer #1: **Yes: **Ebrahim E. Elsayed

---

## [Editor Report · Acceptance letter]

8 Dec 2024

PONE-D-24-39904R1 

PLOS ONE

Dear Dr. Hu, 

I'm pleased to inform you that your manuscript has been deemed suitable for publication in PLOS ONE. Congratulations! Your manuscript is now being handed over to our production team.

Kind regards, 

on behalf of

Mr. Agbotiname Lucky Imoize 

Academic Editor

PLOS ONE